# Synthesis and Memristor Effect of a Forming-Free ZnO Nanocrystalline Films

**DOI:** 10.3390/nano10051007

**Published:** 2020-05-25

**Authors:** Roman V. Tominov, Zakhar E. Vakulov, Vadim I. Avilov, Daniil A. Khakhulin, Aleksandr A. Fedotov, Evgeny G. Zamburg, Vladimir A. Smirnov, Oleg A. Ageev

**Affiliations:** 1Institute of Nanotechnologies, Electronics and Electronic Equipment Engineering, Southern Federal University, 347922 Taganrog, Russia; tominov@sfedu.ru (R.V.T.); avilovvi@sfedu.ru (V.I.A.); dhahulin@sfedu.ru (D.A.K.); aafedotov@sfedu.ru (A.A.F.); ageev@sfedu.ru (O.A.A.); 2Federal Research Centre, The Southern Scientific Centre of the Russian Academy of Sciences, 344006 Rostov-on-Don, Russia; zakhar.vakulov@gmail.com; 3Department of Electrical & Computer Engineering, National University of Singapore, Singapore 117582, Singapore; zamburg@nus.edu.sg

**Keywords:** neuromorphic systems, memristor, ReRAM, resistive switching, nanocrystalline ZnO, pulsed laser deposition, post-growth annealing

## Abstract

We experimentally investigated the effect of post-growth annealing on the morphological, structural, and electrophysical parameters of nanocrystalline ZnO films fabricated by pulsed laser deposition. The influence of post-growth annealing modes on the electroforming voltage and the resistive switching effect in ZnO nanocrystalline films is investigated. We demonstrated that nanocrystalline zinc oxide films, fabricated at certain regimes, show the electroforming-free resistive switching. It was shown, that the forming-free nanocrystalline ZnO film demonstrated a resistive switching effect and switched at a voltage 1.9 ± 0.2 V from 62.42 ± 6.47 (*R_HRS_*) to 0.83 ± 0.06 kΩ (*R_LRS_*). The influence of ZnO surface morphology on the resistive switching effect is experimentally investigated. It was shown, that the ZnO nanocrystalline film exhibits a stable resistive switching effect, which is weakly dependent on its nanoscale structure. The influence of technological parameters on the resistive switching effect in a forming-free ZnO nanocrystalline film is investigated. The results can be used for fabrication of new-generation micro- and nanoelectronics elements, including random resistive memory (ReRAM) elements for neuromorphic structures based on forming-free ZnO nanocrystalline films.

## 1. Introduction

In recent years, the relevance of high-performance computing related to unstructured data classification and pattern recognition has been growing [1,2,3,4,5]. The development of integrated electronics technology leads to an increase in processor performance and an increase in memory volume, but the rate of data exchange between them remains almost unchanged. This issue is common to all computing systems built based on von Neumann architecture, the principle of which is the physical separation of the processor and memory. In this case, the information processing steps are performed sequentially and are limited by the data bus bandwidth (Von Neumann bottleneck) [6,7,8]. It follows that computers based on von Neumann architecture do not efficiently solve problems related to real-time image recognition, diagnostics of various processes, as well as in applications related to self-learning and adaptive control systems. To solve this issue, it is necessary to revise the basic principles of the computer architecture, which would allow one to overcome the Von Neumann bottleneck barrier. The technical implementation of parallelism will allow us to efficiently solve problems associated with multi-parameter nonlinear optimization [9,10], discriminant analysis [11], and clustering methods [12,13] to create systems related to pattern recognition, construction of predictive control systems, and cognitive information processing devices [14].

One of the possible solutions to this problem is the transition of computing systems from the Von Neumann architecture to an architecture close to the structure of the human brain, which is a set of parallel connected low-power computing elements—neurons, interconnected by synapses [15,16]. Like a biological brain, computing systems built on this principle (neuromorphic systems) are adaptive to the environment and resistant to random noise [17]. In such an architecture, data processing is distributed throughout the computing system, rather than concentrated in the central processor. The processor and memory are integrated into a single device, and the information processing steps will be performed in parallel, rather than sequentially. In the general case, a neuromorphic system is a set of layers containing neurons that exchange information with each other using synapses [18]. A feature of a neuromorphic system that allows it to learn how to perform a task is that each connection between neurons has a weight that essentially determines the strength of the signal for the receiving neuron. The weight of each connection can change during operation, allowing one to adjust the operation of the algorithm during training [19,20]. Thus, neuromorphic systems give us the technical ability to implement the work of real synapses and neurons in order to reproduce on their basis the computational primitives of artificial intelligence. It is also worth noting that neuromorphic systems can be integrated into integrated circuits (ICs), expanding the capabilities of complementary metal-oxide-semiconductor (CMOS) technology to a new level of technology [21,22,23]. Neuromorphic systems based on CMOS technology have shown great promise in the development of high-performance systems that can self-learn and adapt to external conditions [24,25,26,27,28]. However, the processing and storage of information using digital-to-analog converters, capacitors, and floating gates, is associated with low accuracy, high energy consumption, and has many design problems, such as voltage changes, noise generation, etc. Additionally, the architecture and communications of neuromorphic systems based on CMOS technology is difficult to scale, which leads to a limitation of the size and functions of the implemented electrical circuits. It follows, that the hardware implementation of neuromorphic systems based on digital integrated circuits is difficult due to several requirements for power consumption, memory size, and the degree of integration. Optimization of existing software algorithms can improve the situation, but not solve it radically. Thus, there is a need to develop novel nanoelectronic elements, which would allow one to realize the functions of biological systems in their interaction with the environment [29].

One of the promising elements for creating artificial intelligence systems is a memristor, the physical implementation of which made it possible to obtain a nanoscale non-volatile device with a large range of resistance values and low power consumption [30,31,32]. The principle of operation of the memristor has an analogy with a biological neuron: a change in conductivity using voltage pulses resembles a change in synaptic weights in brain tissue using chemical impulses [33,34,35]. One of the methods for the technical implementation of neuromorphic systems is the fabrication of memristor structures based on crossbar architecture, which is a set of memristors connected by cross data buses [36,37]. Such a system has a high degree of integration, can perform analog parallel computing, and also can significantly reduce power consumption. A number of requirements are imposed on memory elements for the manufacture of neuromorphic systems, first of all, in terms of the presence of non-volatility properties (the ability to store information in the absence of supply voltage) and analog weight values (the ability to accept three or more stable states), as well as low energy consumption.

Several new technologies for manufacturing memory (MRAM, FeRAM, PRAM, etc.) are presented on the electronic industry market [38,39,40,41,42], one of the most promising of which for creating neuromorphic systems is non-volatile random resistive memory (ReRAM), which has the properties of non-volatility, multi-bit nature, has low power consumption, small cell sizes, high speed, and is also compatible with CMOS technology [43,44]. The ReRAM memory element is structurally a thin film of metal oxide located between two electrodes, and the principle of operation is based on the resistive switching effect (memristive effect), i.e., changes in the resistive state in the range between high (*HRS*) and low (*LRS*) resistance under the influence of an external electric field as a result of the redistribution of oxygen vacancies in the volume of the oxide film. It was demonstrated that many metal oxides (TiO_2_, HfO_2_, NiO, ZrO_2_, and ZnO) exhibit the resistive switching effect [45,46,47,48,49,50], among which nanocrystalline zinc oxide is especially prominent due to high speed, low energy consumption, and a high *HRS/LRS* ratio, which is important for creating ReRAM elements with a large number of intermediate resistive states [51,52,53,54,55,56,57,58].

However, fabrication of ReRAM elements for neuromorphic systems based on nanocrystalline ZnO is currently a difficult task, primarily due to the complexity of controlling the distribution profile and concentration of oxygen vacancies in the oxide film at the stage electroforming. Electroforming is the process of forming oxygen vacancies in the volume of an oxide film under the influence of an external electric field, which is thermal breakdown. Therefore, in recent years, the attention of scientific communities has been directed to the search for nanomaterials, as well as the modes of their fabrication, which would exhibit the resistive switching effect without an additional stage of electroforming (forming-free memristors) [59,60,61,62]. Currently, the following technological methods are widely used in the fabrication of ZnO nanocrystalline films: magnetron sputtering [63], chemical vapor deposition [64], sol–gel process [65], anodic oxidation [66], thermal evaporation [67], atomic layer deposition (ALD) [68], and pulsed laser deposition (PLD) [69,70]. The latter has shown its promise for fabrication of metal oxide films, since it allows one to control a large number of technological parameters, which makes it possible to widely influence the electrophysical, physicochemical, mechanical, and structural parameters of the films [71]. In particular, the additional process of annealing after oxide growth affects the concentration of oxygen vacancies in the film [72]. The purpose of this work is to study the effect of post-growth annealing modes on the morphological, structural, and electrophysical parameters of ZnO nanocrystalline films obtained by the PLD, as well as to study the effect of post-growth annealing modes on the forming-free and the resistive switching effect in ZnO nanocrystalline films.

## 2. Materials and Methods

The experimental samples were fabricated using a Pioneer 180 pulsed laser deposition (PLD) system (Neocera Co., Beltsville, MD, USA) equipped with a KrF excimer laser with a wavelength of 248 nm and an energy of 200 mJ. Al_2_O_3_ plates with crystallographic orientation (0001), previously purified in acetone and isopropyl alcohol at temperatures of 56 °C and 83 °C, respectively, were used as a substrate. As the lower contact layer, a TiN film 50 ± 7 nm thick was used, formed by the PLD method under the following conditions: substrate temperature 600 °C, number of pulses 12,000, frequency 10 Hz, argon pressure 1 Torr. Then, a nanocrystalline ZnO film was deposited on top of TiN under the following conditions: substrate temperature 800 °C, number of pulses 60,000, laser pulse repetition rate 10 Hz, oxygen pressure 10^−3^ Torr. To provide electrical contact to the lower electrode, the deposition of nanocrystalline ZnO was carried out using a special shadow mask. As a result, 5 samples Al_2_O_3_/TiN/ZnO were prepared, which were annealed in nitrogen atmosphere with a pressure of 10^−3^ Torr at annealing temperatures of 25 °C (i.e., without annealing), 600 °C, 800 °C, 1000 °C, and 1200 °C for 10 h each.

The morphology of ZnO nanocrystalline films was studied by atomic force microscopy (AFM) in the semi-contact mode using the Ntegra Probe Nanolaboratory (NT-MDT, Zelenograd, Russia) and a commercial cantilever an NSG11 with 255 kHz resonant frequency and 11.8 N/m spring constant. The ZnO thickness was determined from the ZnO/Al_2_O_3_ interface measuring. Based on the results obtained, the dependences of surface roughness and grain diameter of ZnO nanocrystalline films on the annealing temperature were plotted. To investigate surface conductivity of nanocrystalline ZnO, conductive AFM and Kelvin Probe modes were used. For this purpose, a commercial ETALON HA_HR cantilevers with a conductive coating of W_2_C, 380 kHz resonant frequency and 34 N/m spring constant were used. Processing of the results was carried out using the NT-MDT software package «Image Analysis 2.0».

The structure of ZnO films was studied by X-ray photoelectron spectroscopy using an ESCALAB 250Xi spectrometer (Thermo Scientific, Waltham, MA, USA) combined spectrometer with monochromatization of the Al Kα X-ray radiation line. The energy resolution was determined in reference to the Ag 3d5/2 line and corresponded to 0.6 eV. In the study, the spatial resolution was 250 μm. Additionally, the structure of ZnO films was studied using X-ray diffractometry using a Rigaku Miniflex 600 diffractometer (Rigaku Corporation, Tokyo, Japan). 

The electrophysical parameters of ZnO nanocrystalline films were studied using an Ecopia HMS-3000 Hall effect system (Ecopia Co., Anyang, Korea). The dependences of electron concentration, electron mobility, and resistivity of ZnO nanocrystalline films on the annealing temperature were plotted.

To study the effect of annealing time on roughness and resistivity, 4 samples of Al_2_O_3_/TiN/ZnO were fabricated and obtained at different substrate temperatures of 300 °C and 800 °C. Then the films were annealed in a nitrogen atmosphere at pressures of 10^−1^ Torr and 10^−3^ Torr and a temperature of 1200 °C for 1, 5, and 10 h. As a result, four ZnO films were obtained, grown, and annealed at various substrate temperatures and nitrogen pressure: 300 °C and 10^−1^ Torr, 300 °C and 10^−3^ Torr, 800 °C and 10^−1^ Torr, and 800 °C and 10^−3^ Torr. Based on the results obtained, the dependences of surface roughness and specific resistance of ZnO nanocrystalline films on the annealing time were plotted.

The resistive switching effect in ZnO nanocrystalline films was studied under atmospheric conditions using a Keithley 4200-SCS semiconductor parameter analyzer (Keithley Instruments, Solon, OH, USA) and an EM-6070A submicron probe system (Planar, Minsk, Belarus). The TiN film served as the lower contact; a tungsten probe with diameter 150 nm was used as the upper contact. In order to avoid thermal breakdown of ZnO nanocrystalline films, the compliance current was set to 1 mA.

To determine the electroforming voltage (*U_e_*) of all experimental samples, an Al_2_O_3_/TiN/ZnO/W structure was supplied with voltage in the form of a continuous linear sweep in the range from 0 to 18 V and a rise rate of 2 V/s. A drastic increase in current was observed in each sample at a certain voltage value, depending on the annealing time of the ZnO nanocrystalline film. In this case, the experiment was stopped, and the resulting voltage was taken as *U_e_*. According to the results obtained, the dependence of the electroforming voltage on the grain size was built.

To study the effect of the grain’s diameter on the resistance of a ZnO nanocrystalline film on each sample, current–voltage (*I–V*) characteristics were obtained at 40 points on the ZnO surface. For this, an alternating sawtooth sweep voltage was applied to the Al_2_O_3_/TiN/ZnO/W structure for 8 s and an amplitude of 2–8 V. The read voltage was 0.5 V. Based on the obtained experimental results, the dependences of the resistances *R_HRS_* and *R_LRS_*, as well as the ratios *R_HRS_/R_LRS_* on the grain diameter were built.

The study of the influence of the electroforming voltage and the electroforming time on the resistive switching effect was carried out in several stages. At the first stage, the sample was subjected to electroforming at a point on the surface of the ZnO nanocrystalline film at a direct sweep voltage with an amplitude *U_e_* determined earlier for each sample. At the second stage, the current–voltage characteristic was measured at this point. Then, the electroforming voltage for each sample increased by 1 V and the study was repeated at another point on the surface of the ZnO nanocrystalline film. Based on the obtained experimental results, the dependences of the *R_HRS_/R_LRS_* ratio and switching voltage (*U_set_*) on the *U_e_* at an electroforming time of 1 s were built for each sample, as well as the dependences of the *R_HRS_/R_LRS_* and *U_set_* on the electroforming time (*t_e_*) at a voltage *U_e_*.

To investigate the effect of the forming-free ZnO nanocrystalline film morphology on the occurrence of the resistive switching effect, we used a conductive AFM technique, which was described in [54]. For this purpose, two regions were scanned on the ZnO surface at voltages of different polarity. The first region was formed by scanning a surface of 10 × 10 μm^2^ with a voltage of −3 V applied to the probe. To create a second region inside the scan 10 × 10 μm^2^, a surface scan of 6 × 6 μm^2^ was applied at a voltage of +3 V. After the creation of regions according to the described technique, the surface of the ZnO nanocrystalline film was studied in the semi-contact AFM mode, in the spreading resistance mode with a constant voltage +1 V, and in the Kelvin Probe mode.

To investigate the uniformity and homogeneity of the resistive switching effect, 300 *I–V* characteristics at one point, as well as 50 *I–V* characteristics at different points, were obtained on the surface of a ZnO nanocrystalline film. Based on the results obtained, the dependences of *R_HRS_* and *R_LRS_* on the cycle number and point number on the surface of the ZnO film were plotted.

To study the influence of control parameters on the resistive switching effect in the forming-free ZnO nanocrystalline film, *I–V* characteristics were obtained at the compliance current in the range from 0.1 to 5.0 mA, and also with a voltage amplitude *U_A_* in the range from 2 to 6 V. Based on the obtained experimental results, the dependences of the *R_HRS_/R_LRS_* ratio on the compliance current, as well as on *U_A_*, were plotted.

## 3. Results and Discussion

Figure 1 shows AFM images of ZnO nanocrystalline films after annealing at various temperatures. An analysis of the results of measuring the ZnO/Al_2_O_3_ interface before annealing showed that the film thickness was 71.3 ± 35.4 nm (Figure 1f).

Based on the analysis of the AFM images, the dependences of the surface roughness and grain diameter of ZnO nanocrystalline films on the annealing temperature were plotted, shown in Figure 2. It was found that with increasing annealing temperature, the surface roughness decreased from 35.0 ± 4.1 to 13.1 ± 1.8 nm, and the grain diameter increased from 231 ± 12 to 511 ± 35 nm. A significant increase in grain diameter was observed in films annealed at temperatures above 600 °C. This effect can be associated with the intensification of the coalescence process and the subsequent formation of a coarser-grain structure in the films. At a high temperature, the atoms have enough activation energy for the internal diffusion process to complete the crystal lattice. Thus, with an increase in the annealing temperature, the grain size with lower surface energy increased (Figure 2).

To investigate the elemental composition and crystal structure, we selected samples fabricated at 1200 °C, since the samples possess the lowest roughness in comparison with other samples, which is important for the manufacture of ReRAM elements with reproducible parameters. Figure 3a shows the XPS spectra of ZnO films before and after annealing. The spectra showed lines corresponding to C1s (284.6 eV), O1s (530.8 eV), Zn 2p 3/2 (1022 eV), Zn 2p 1/2 (1045 eV), and N 1s (400 eV). Under the influence of annealing in nitrogen atmosphere, the intensity of O 1s decreased. The decrease in oxygen concentration for the sample annealed in N_2_ occurred due to weak oxygen absorption. Annealing in N_2_ led to a deviation from the stoichiometric state, which led to a lack of oxygen in ZnO films, in contrast to annealing in some oxygen. In this case the concentration of oxygen vacancies decreased due to chemisorption, which allowed oxygen to interact with a sufficient number of Zn atoms to form ZnO [54].

Figure 3b shows the XRD spectrum of a ZnO film without annealing. It was characterized by a pronounced peak of ZnO (002), which was observed at 34°. In addition, peaks of ZnO (002) and ZnO (004) were also observed at 38° and 72°.

Figure 4 shows the dependences of the electrophysical parameters of ZnO films on the annealing temperature. As the annealing temperature increased from 25 to 1200 °C, an increase in the electron concentration in the films was observed from (5.1 ± 1.4) × 10^13^ to (2.2 ± 0.3) × 10^15^ cm^−3^, while the electron mobility decreased from 19 ± 2 to 13 ± 1 cm^2^/V·s. This effect may be associated with a change in the stoichiometric composition of the films due to a release of oxygen during annealing in a nitrogen atmosphere, which was confirmed by the results of XPS studies, and, in turn, caused a decrease in the film resistivity from 93.12 ± 14.52 to 7.32 ± 3.21 Ω·cm (the concentration of oxygen vacancies increased).

Studies of the effect of annealing duration on the morphological and electrical parameters of films at an annealing temperature of 1200 °C showed that, with an increase in the annealing duration, surface roughness and film resistivity decreased (Figure 5). According to the results of experimental studies, an increase in the nitrogen pressure in the chamber during annealing of the films from 10^−3^ Torr to 10^−1^ Torr led to a slight decrease in the surface roughness of the films and resistivity. An increase in the substrate temperature also led to a decrease in the surface roughness of the films and the resistivity. To study the resistive switching effect, from samples obtained at an annealing temperature of 1200 °C, we selected a film obtained at a substrate temperature of 800 °C, a nitrogen pressure of 10^−1^ Torr and annealed for 10 h, since this one had the smallest roughness and the lowest resistivity.

An analysis of the experimental results showed that an increase in grain size from 231 ± 12 to 511 ± 35 nm led to a decrease in the electroforming voltage from 14.1 ± 2.7 to 0.3 ± 0.2 V (Figure 6). This effect can be explained by an increase in the concentration of oxygen vacancies at the boundaries and in the bulk of grains, as well as an increase in the electron concentration (Figure 4a) with an increase in the annealing time, which leads to an increase in the conductivity of the ZnO nanocrystalline film. At the same time, in ZnO films annealed at temperatures ranging from 25 to 1000 °C, the concentration of oxygen vacancies turned out to be insufficient so that without carrying out the electroforming operation, the film resistance can be switched at voltage values commensurate with the switching voltage *U_set_*, which is usually units of volts. On the other hand, the electroforming voltage *U_e_* of the ZnO nanocrystalline film annealed at a temperature of 1200 °C is lower than the switching voltage *U_set_*. This means that this film does not require an electroforming step. A decrease in the dispersion with a decrease in *U_e_* is most likely associated with a decrease in the Joule heat release during thermal breakdown during the flow of the current through the ZnO film.

Analysis of the *I–V* characteristics showed that the ZnO films on which the electroforming operation was carried out had a linear bipolar switching of the resistance (Figure 7a). According to the literature, in this case, the kinetic energy of electrons is the dominant parameter among three other physical parameters: the gradient of the electric field, the concentration gradient of oxygen vacancies, and the temperature gradient [73]. This type of switching corresponds to the filament mechanism of the resistive switching effect, based on the generation of a nanoscale conduction channel from oxygen vacancies under the influence of the electroforming voltage. In this case, the nanoscale conduction channel in the cross section was a cone (insert in Figure 7a). Taking into account the fact that electroforming was carried out by applying a positive bias to the W electrode, and oxygen vacancies have a conditionally positive charge, we could assume that the base of the nanoscale conduction channel is located at the TiN/ZnO interface, and the top at the ZnO/W interface. Moreover, the presence of ohmic conductivity on the current–voltage characteristic of the TiN/ZnO/W structure can be explained by the connection of the upper and lower contacts with a nanoscale conduction channel (Figure 7a). According to the filament mechanism, switching the resistance of the oxide film consists of changing the diameter of the conduction channel top by generating and destroying oxygen vacancies under the influence of an external electric field. Thus, it was shown that the ZnO film without annealing (25 °C for 10 h in N_2_) after the operation of electroforming switched from the state of high resistance *R_HRS_* to the low resistance state *R_LRS_* at a voltage of 2.7 ± 0.2 V, and at a voltage of −2.9 ± 0.1 V switched back to *R_HRS_*.

An analysis of the experimental results showed that the *I–V* characteristic of the ZnO nanocrystalline film that does not require electroforming exhibited a nonlinear bipolar switching of the resistance (Figure 7b). According to the literature, in this case, the electric field gradient is the dominant parameter [73]. This type of switching corresponds to the barrier mechanism of the resistive switching effect based on the change in resistance as a result of the redistribution of the concentration profile of oxygen vacancies over the volume of the oxide film under an external electric field (insert in Figure 7). In this case, oxygen vacancies were formed over the volume of the oxide film, both at the boundaries and inside the grains, in contrast to the filamentary mechanism in which nanoscale conduction channels were formed mainly at the grain boundaries. According to the barrier mechanism, the volume of the oxide film was divided into two sections with different resistance *R_1_* < *R_2_*, the length of which depends on the distribution of the profile of oxygen vacancies controlled by an external electric field (insert in Figure 7b). Thus, it was shown that a ZnO film annealed at a temperature of 1200 °C for 10 h, without performing an electroforming operation, switched from a state of high resistance *R_HRS_* to a state of low resistance *R_LRS_* at a voltage of 1.9 ± 0.2 V, and at a voltage of −1.4 ± 0.5 B returned to *R_HRS_* state.

An analysis of the experimental results showed that an increase in grain diameter from 231 ± 12 to 511 ± 35 nm led to an increase in *R_HRS_* resistance from 20.54 ± 8.22 to 62.42 ± 6.47 kΩ, and a decrease in *R_LRS_* resistance from 0.91 ± 0.30 to 0.83 ± 0.06 kΩ (Figure 8a), and, as a result, to increase the *R_HRS_*/*R_LRS_* ratio from 24.61 ± 16.12 to 78.34 ± 5.31 (Figure 8b). From Figure 4c it followed that the resistivity decreased with increasing grain diameter, but *R_HRS_* in Figure 8a increased. This can be explained by a decrease in the resistance of the ZnO nanocrystalline film after electroforming. On the other hand, the resistance of the *R_LRS_* changed slightly with increasing grain diameter. In the case of the filament mechanism, this may be due to an increase in the top diameter of the nanoscale conduction channel to the diameter of the probe/ZnO contact upon the transition of the films to the *LRS* state. In the case of the barrier mechanism this may be due to a uniform redistribution of oxygen vacancies over the volume of the oxide film.

An analysis of the experimental results showed that an increase in *U_e_* in the range from 1 to 20 V led to a decrease in the *R_HRS_/R_LRS_* ratio in the range from 78.34 ± 5.31 to 12.3 ± 3.2 (Figure 9a). This can be explained by an increase in the concentration of oxygen vacancies in the volume of the ZnO film due to the release of Joule heat with increasing *U_e_*, and, therefore, to a decrease in the *R_HRS_* resistance. In the dependences corresponding to ZnO films with grain diameters from 511 ± 35 to 347 ± 28 nm, two sections can be conditionally distinguished, on one of which a gradual decrease in the *R_HRS_/R_LRS_* ratio was observed, which corresponds to controlled thermal breakdown, and on the other, a drastically decrease in the ratio *R_HRS_/R_LRS_*, which corresponds to uncontrolled thermal breakdown, in which the current–voltage characteristic of the TiN/ZnO/W structure began to degenerate into a linear dependence. A decrease in the dispersion in the region with uncontrolled thermal breakdown may be due to the achievement of the maximum possible concentration of oxygen vacancies in the investigated region of the ZnO film at sufficiently high *U_e_* voltages.

An analysis of the experimental results showed that an increase in the electroforming time *t_e_* in the range from 1 to 2 s led to an increase in the *R_HRS_/R_LRS_* ratio in the range from 23.12 ± 9.31 to 80.21 ± 8.64 (Figure 9b) for ZnO films with grain diameters in the range from 231 ± 12 to 421 ± 32 nm, which can be explained by the generation of oxygen vacancies and the formation of a nanoscale conduction channel.

A further increase in the electroforming time from 2 to 8 s led to a decrease in the *R_HRS_/R_LRS_* ratio in the range from 2.11 ± 0.65 to 8.21 ± 1.33, which could also be associated with a further increase in the concentration of oxygen vacancies, an increase in the diameter of the nanoscale conduction channel top in the *HRS* state, and as a result, a decrease in resistance *R_HRS_*. An analysis of the results showed, that an increase in the electroforming time in the range from 1 to 8 s led to a decrease in the *R_HRS_/R_LRS_* ratio from 78.34 ± 5.31 to 12.31 ± 1.03 for a film with a grain diameter of 511 ± 35 nm, which was also explained by the generation of excess oxygen vacancies in the bulk of the ZnO film.

An analysis of the experimental results showed that an increase in *U_e_* in the range from 1 to 20 V led to an increase in the switching voltage *U_set_* in the range from 1.9 ± 0.2 to 4.5 ± 0.3 V (Figure 10a). Increase in the electroforming time *t_e_* in the range from 1 to 5 s led to an increase in the switching voltage *U_set_* in the range from 1.9 ± 0.2 to 6.4 ± 0.4 V (Figure 10b). This may be due to a decrease in the resistance of the ZnO film due to an increase in the concentration of oxygen vacancies with increasing *U_e_* and *t_e_*, and as a result, a decrease in the Joule heat, that is necessary for changing the resistance state at the same voltage values.

An analysis of the 10 × 10 μm^2^ and 6 × 6 μm^2^ regions obtained by conductive AFM on the ZnO film surface (Figure 11) at voltages of −3 and 3 V, respectively, showed the presence on the surface of a region in the *HRS* state with a resistance of (8.3 ± 1.2) × 10^−9^ Ω (dark contrast), and region in the *LRS* state with resistance (2.1 ± 0.6) × 10^−9^ Ω (light contrast). Analysis of the Kelvin mode results (Figure 11c) showed that a higher surface potential was obtained by applying a negative potential to the probe (Figure 11d), and a region with a high resistance was formed on which charge accumulation occurred. When a region with a low resistance was formed, the value of the surface potential decreased, which correlates well with the literature data [54,74]. Thus, it was shown that the ZnO nanocrystalline film that does not require electroforming exhibited a stable resistive switching effect, which was weakly dependent on its nanoscale structure.

An analysis of the experimental results of studying the uniformity of the resistive switching effect (Figure 12a) showed that *R_HRS_* and *R_LRS_* were 66.28 ± 7.73 kΩ and 0.83 ± 0.14 kΩ, respectively. An increase in the dispersion of the *R_HRS_* and *R_LRS_* values with an increase in the number of measurements can be associated with an increase in the concentration of oxygen vacancies due to repeated switching of the resistive state of ZnO, which leads to an increase in the uneven distribution of the profile of oxygen vacancies over the volume of the oxide film.

An analysis of the experimental results of studying the homogeneity (Figure 12b) showed that *R_HRS_* and *R_LRS_* were 66.22 ± 12.18 kΩ and 0.87 ± 0.43 kΩ. High dispersion of *R_HRS_* and *R_LRS_* can be associated with morphological inhomogeneity of the probe/sample contact at different points on the surface of the ZnO film.

Studying the influence of control parameters on the resistive switching effect in an electroforming-free ZnO nanocrystalline film showed that an increase in the compliance current from 0.1 to 1 mA led to an increase in the *R_HRS_*/*R_LRS_* ratio from 53.12 ± 5.17 to 78.34 ± 5.31 (Figure 13a). Increase in the amplitude of the sweep voltage *U_A_* from 2 to 4 V led to an increase in the *R_HRS_*/*R_LRS_* ratio from 63.87 ± 6.72 to 78.34 ± 5.31 (Figure 13b). This can be explained by the increase in the distance that oxygen vacancies travel towards the top electrode with an increase in the compliance current, which led to a decrease in the *R_LRS_* (insert in Figure 7b). A further increase in the compliance current from 1 to 5 mA led to a decrease in the *R_HRS_*/*R_LRS_* ratio from 78.34 ± 5.31 to 17.73 ± 9.26. Increase in *U_A_* from 4 to 6 V led to a decrease in the *R_HRS_*/*R_LRS_* ratio from 78.34 ± 5.31 to 25.61 ± 4.25. This may be due to a decrease in the difference between the resistances *R_1_* and *R_2_* in the bulk of the oxide film (insert in Figure 7b) due to an increase in the concentration of oxygen vacancies, which led to a decrease in *R_HRS_*.

As a result, we determined the regimes of nanocrystalline ZnO films manufacturing exhibiting a stable memristive effect with a *R_HRS_/R_LRS_* ratio 78.34 ± 5.31 and a switching voltage *U_SET_* 1.9 ± 0.2 V. The absence of the need for an electroforming allows one to avoid a number of problems with the ReRAM element efficiency, associated with information loss due to reducing the weight of synapses between neurons, and also allows us to increase the output of workable ReRAM elements. Thus, the results of the study, in conjunction with the results of other scientific groups studying the use of zinc oxide for biological synapses [75,76,77], show us the prospects of using ZnO nanocrystalline films for neuromorphic systems manufacturing.

## 4. Conclusions

The paper presented the results of experimental studies of the effect of post-growth annealing on the morphological, structural, and electrophysical parameters of nanocrystalline ZnO films fabricated by pulsed laser deposition. It was demonstrated that increasing the annealing temperature from 25 to 1200 °C led to a decrease in the film roughness from 35.0 ± 4.1 to 13.1 ± 1.8 nm, and to an increase in grain size from 231 ± 12 to 511 ± 35 nm. It was also shown that post-growth annealing in a nitrogen atmosphere led to a decrease in the oxygen concentration in ZnO nanocrystalline films. An analysis of the experimental results of the effect of annealing temperature *t_A_* on electrophysical parameters showed that an increase in the *t_A_* from 25 to 1200 °C led to an increase in the electron concentration in the films from (5.1 ± 1.4) × 10^13^ to (2.2 ± 0.3) × 10^15^ cm^−3^, a decrease in electron mobility from 19 ± 2 to 13 ± 1 cm^2^/V·s, as well as a decrease in the film resistivity from 93.12 ± 14.52 to 7.32 ± 3.21 Ω·cm, which is explained by an increase the concentration of oxygen vacancies in the bulk of the ZnO film. It was shown that an increase in annealing time and temperature, as well as an increase in nitrogen pressure, leading to a decrease in surface roughness and a decrease in the resistivity of ZnO films.

An analysis of the experimental results showed that an increase in grain diameter from 231 ± 12 to 511 ± 35 nm led to a decrease in the electroforming voltage from 14.1 ± 2.7 to 0.3 ± 0.2 V. It was shown that the ZnO nanocrystalline film annealed at a temperature of 1200 °C for 10 h exhibited a resistive switching effect without an additional electroforming operation, switched from *R_HRS_* to *R_LRS_* at a voltage of 1.9 ± 0.2 V, and switched at a voltage of −1.4 ± 0.5 V back from *R_LRS_* to *R_HRS_*.

It was shown that an increase in grain size from 231 ± 12 to 511 ± 35 nm led to an increase in *R_HRS_* resistance from 20.54 ± 8.22 to 62.42 ± 6.47 kΩ, a decrease in *R_LRS_* resistance from 0.91 ± 0.30 to 0.83 ± 0.06 kΩ, and also to an increase in the ratio *R_HRS_*/*R_LRS_* from 24.61 ± 16.12 to 78.34 ± 5.31. An increase in *U_e_* in the range from 1 to 20 V led to a decrease in the *R_HRS_*/*R_LRS_* ratio in the range from 78.34 ± 5.31 to 12.3 ± 3.2.

An increase in the electroforming time *t_e_* in the range from 1 to 2 s led to an increase in *R_HRS_*/*R_LRS_*, and from 2 to 8 s led to a decrease in *R_HRS_*/*R_LRS_*, which may be associated with an increase in the diameter of the nanoscale conduction channel top in the *HRS* state, and, as a result, decrease in resistance *R_HRS_*. An analysis of the experimental results showed that an increase in *U_e_* in the range from 1 to 20 V led to an increase in the switching voltage *U_set_* in the range from 1.9 ± 0.2 V to 4.5 ± 0.3 V, and an increase in the electroforming time *t_e_* in the range from 1 to 5 s led to an increase in voltage *U_set_* switching in the range from 1.9 ± 0.2 to 6.4 ± 0.4 V.

An analysis of the experimental results of studying the effect of morphology on the resistive switching effect showed that the ZnO nanocrystalline film exhibited a stable resistive switching effect, which is weakly dependent on its nanoscale structure.

An analysis of the experimental results of studying the uniformity of the resistive switching effect showed that *R_HRS_* and *R_LRS_* were 66.28 ± 7.73 kΩ and 0.83 ± 0.14 kΩ. A study of the homogeneity of the resistive switching effect showed that *R_HRS_* and *R_LRS_* were 66.22 ± 12.18 kΩ and 0.87 ± 0.43 kΩ.

Investigation of the control parameters influence on the resistive switching effect in an electroforming-free ZnO nanocrystalline film showed that an increase in the compliance current from 0.1 to 1 mA led to an increase in the *R_HRS_*/*R_LRS_* ratio from 53.12 ± 5.17 to 78.34 ± 5.31. An increase in the amplitude of the scan voltage *U_A_* from 2 to 4 V led to an increase in the ratio *R_HRS_*/*R_LRS_* from 63.87 ± 6.72 to 78.34 ± 5.31.

The results can be used in the manufacture of new-generation micro- and nanoelectronics elements to create ReRAM elements of neuromorphic structures based on forming-free ZnO nanocrystalline films.

## Figures and Tables

**Figure 1 nanomaterials-10-01007-f001:**
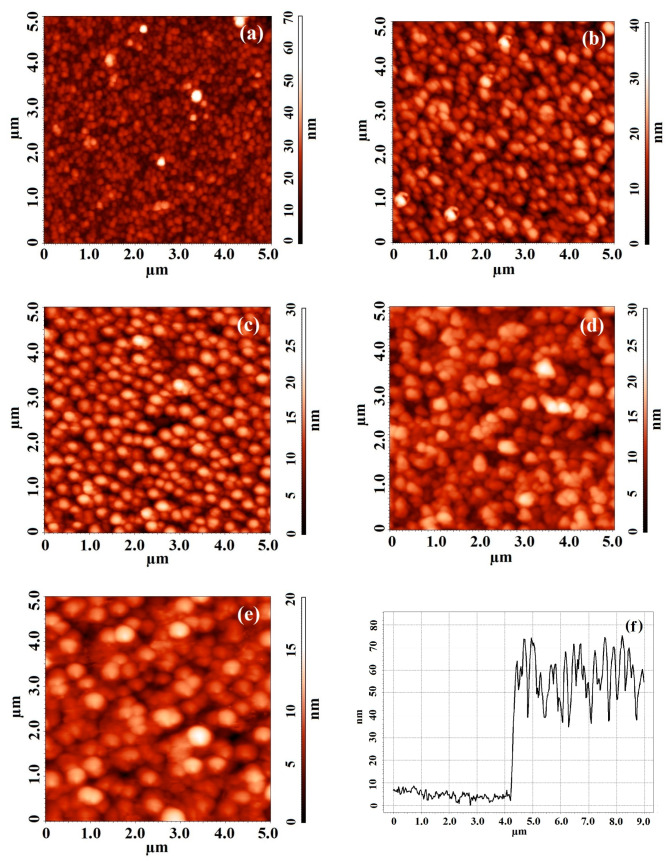
AFM images of nanocrystalline ZnO films fabricated at various annealing temperatures: (**a**)—without annealing (25 °C); (**b**)—600 °C; (**c**)—800 °C; (**d**)—1000 °C; (**e**)—1200 °C, and (**f**)—AFM cross-section of the ZnO/Al_2_O_3_ interface before annealing.

**Figure 2 nanomaterials-10-01007-f002:**
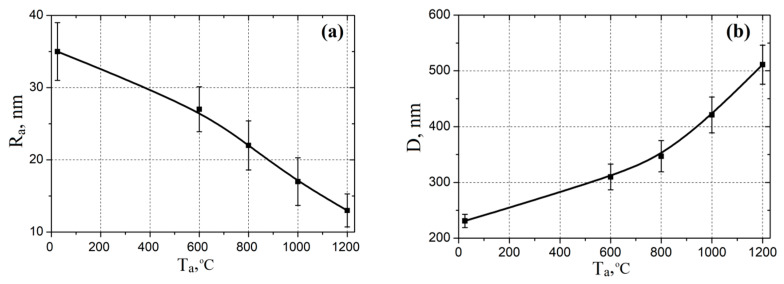
Experimental investigation of annealing temperature effect on the surface roughness (**a**) and grain’s diameter (**b**).

**Figure 3 nanomaterials-10-01007-f003:**
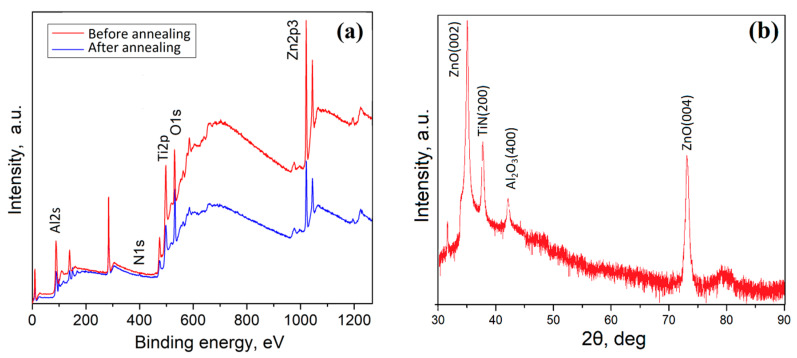
XPS (**a**) and XRD (**b**) spectra of the nanocrystalline ZnO film.

**Figure 4 nanomaterials-10-01007-f004:**
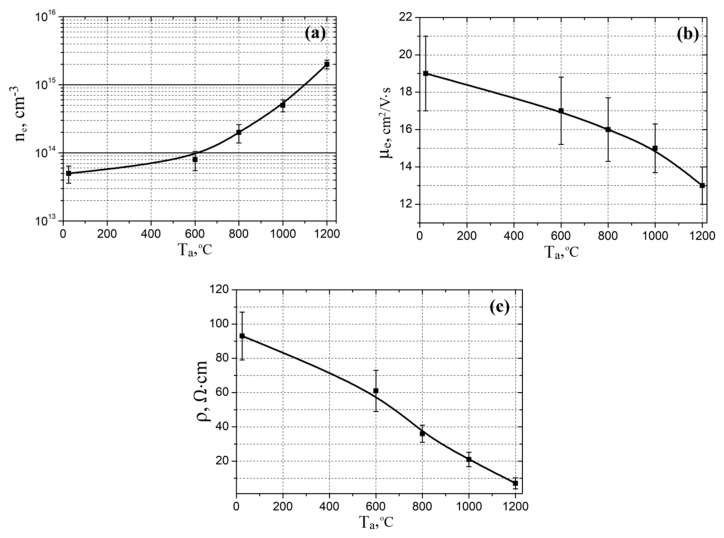
Experimental investigation of annealing temperature effect on electron concentration (**a**), electron mobility (**b**), and resistivity (**c**).

**Figure 5 nanomaterials-10-01007-f005:**
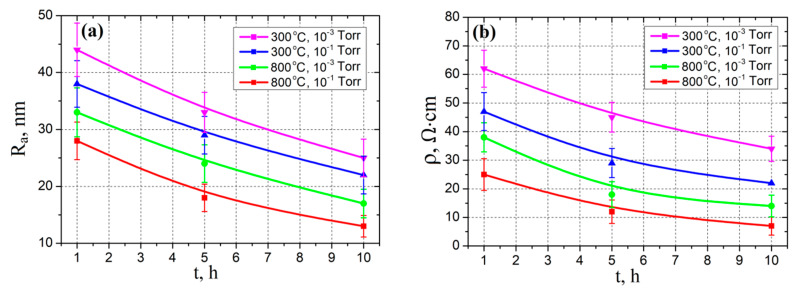
Experimental investigation of annealing time effect on surface roughness (**a**) and resistivity (**b**) at different substrate temperatures and chamber pressures.

**Figure 6 nanomaterials-10-01007-f006:**
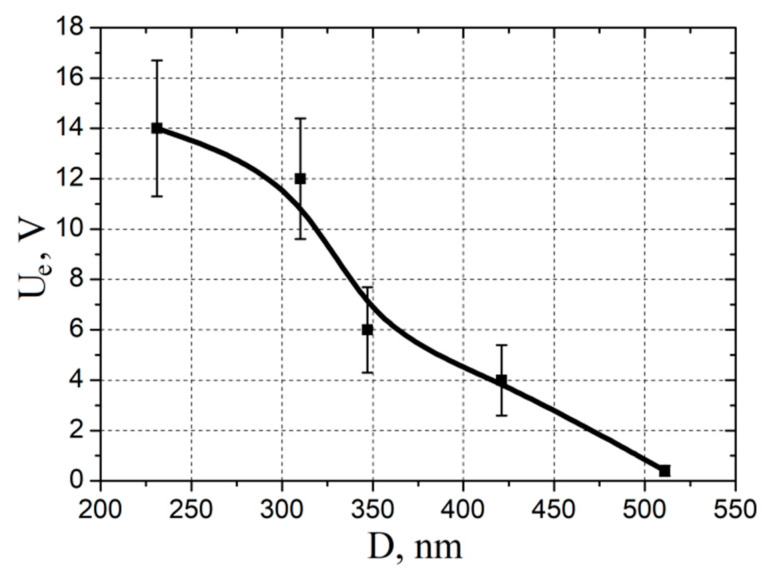
Experimental investigation of the electroforming voltage effect on the grain diameter.

**Figure 7 nanomaterials-10-01007-f007:**
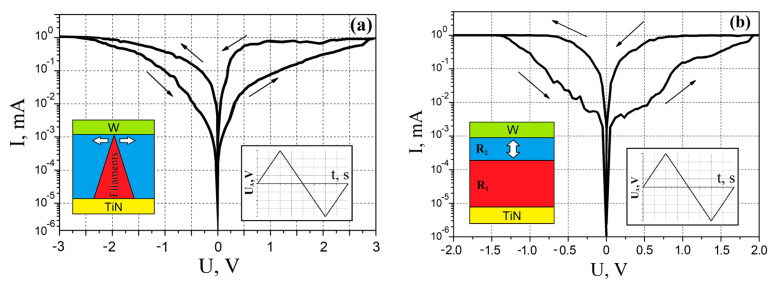
*I–V* characteristics of nanocrystalline ZnO films: (**a**)—without annealing (25 °C) and (**b**)—after annealing at 1200 °C.

**Figure 8 nanomaterials-10-01007-f008:**
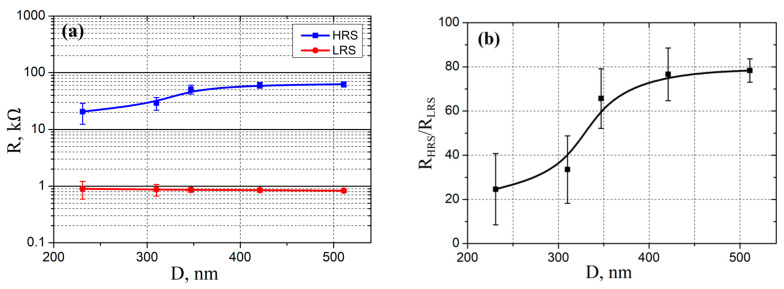
Experimental investigation of grain diameter effect on: (**a**)—nanocrystalline ZnO film resistance and (**b**)—*R_HRS_*/*R_LRS_* ratio.

**Figure 9 nanomaterials-10-01007-f009:**
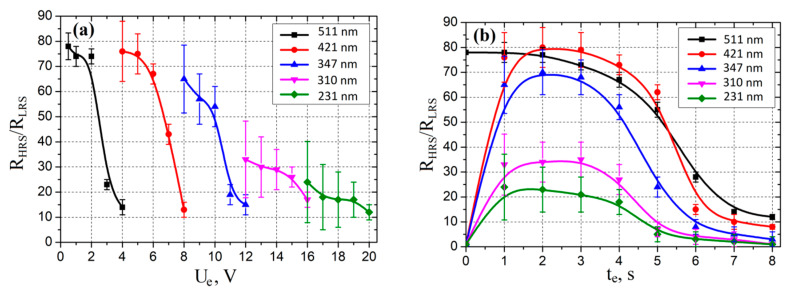
Experimental investigations of *R_HRS_*/*R_LRS_* ratio dependences on electroforming voltage (**a**) and electroforming time (**b**) at various grain diameters.

**Figure 10 nanomaterials-10-01007-f010:**
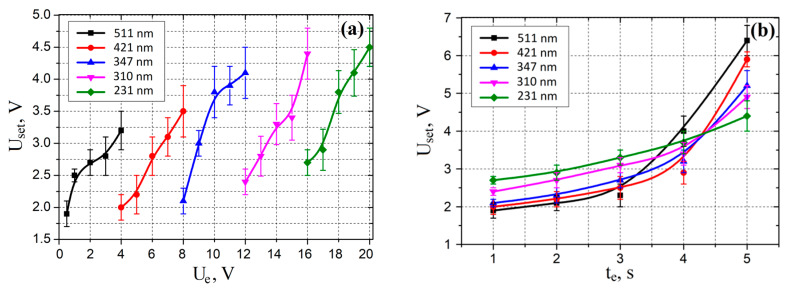
Experimental investigations of set voltage dependences on electroforming voltage (**a**) and electroforming time (**b**) at various grain diameters.

**Figure 11 nanomaterials-10-01007-f011:**
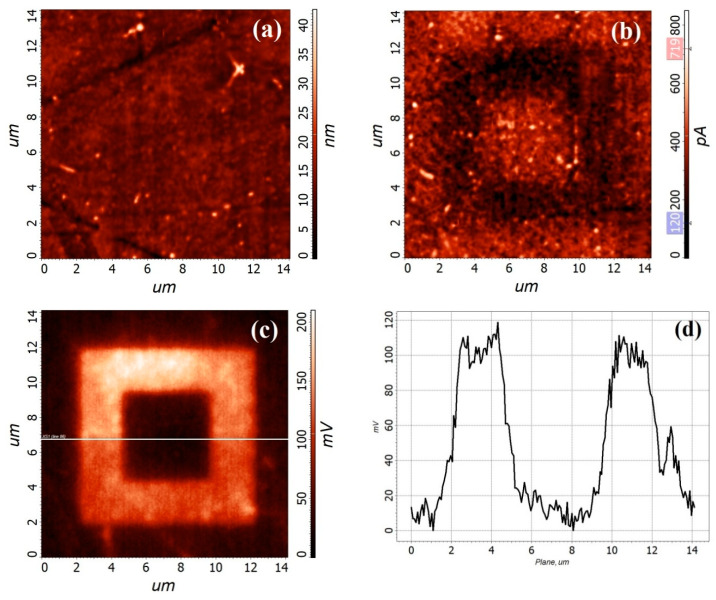
AFM image of the electroforming-free nanocrystalline ZnO film surface: (**a**)—morphology, (**b**)—current contrast, (**c**)—potential distribution, and (**d**)—profilogram along the line in (**c**).

**Figure 12 nanomaterials-10-01007-f012:**
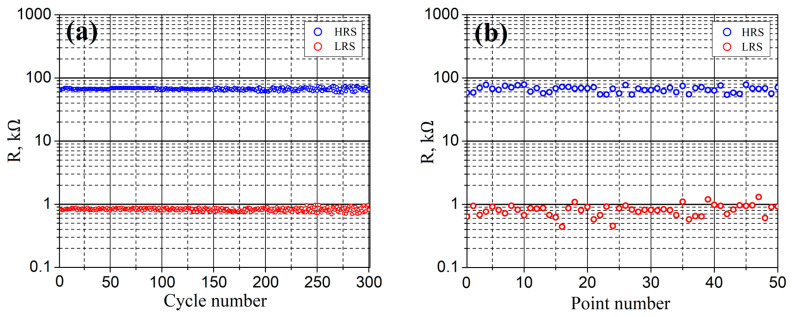
Investigation of resistive switching in electroforming-free nanocrystalline ZnO film: (**a**)—uniformity and (**b**)—homogeneity.

**Figure 13 nanomaterials-10-01007-f013:**
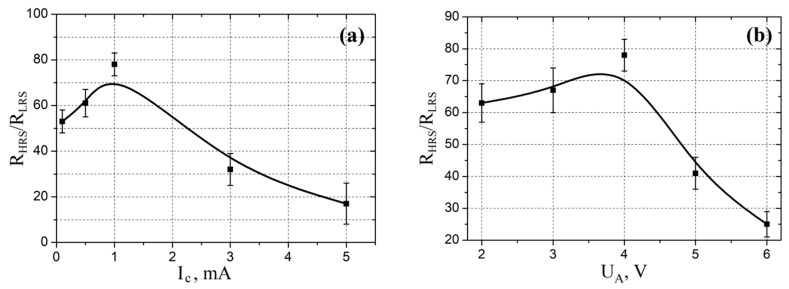
Experimental investigations of dependences of *R_HRS_*/*R_LRS_* ratio on (**a**) current compliance and (**b**) sweep voltage.

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
