# Peer review of "Synthesis and Memristor Effect of a Forming-Free ZnO Nanocrystalline Films"

_nanomaterials, 2020, doi:10.3390/nano10051007_

Round 1
Reviewer 1 Report
Dear Authors,
I have revised the manuscript “Synthesis and memristor effect of a forming-free
ZnO nanocrystalline films” by Roman V. Tominov” and co-workers, submitted for publication in NANOMATERIALS.
The manuscript reports an experimental study of ZnO nanocrystalline film deposited by PVD and subjected to annealing with different process parameters to investigate the resistive switching behaviour of the material.
The manuscript is well written, with detailed description of the experimental equipment and procedure used, and rigorous in data exposure.
Some minor issues are present and a major one concerning AFM morphological measurements which may affects the manuscript in many parts.
Here in the following my general and specific comments. Specific requests are also given in the ANNOTATED VERSION OF THE MANUSCRIPT I have attached to with my Reviewer Response File.
COMMENTS TO AUTHORS
- The Figure 1 shows AFM images of ZnO samples morphology. What is apparent is the probable presence of artefacts in the images due to the worn tip. More specifically the features that indicate the presence of artifacts in the images are the triangular features with the same orientation which are present in all the different images (in one of them the features has a trapezoidal shape with same orientation of the borders). The Authors should check if by rotating the sample respect to the cantilever the features remain the same (artifacts) or rotates (actual features of the sample morphology). If those reported in the manuscript are not the actual sample morphologies the authors should show (replace) additional data and discuss the experimental findings accordingly. It is likely that the trends observed in roughness and grain diameter are actual, but the absolute may be different.
As a consequence data and discussion which refers to the grain diameter which appear in several parts of the manuscript may require amendment. I have highlighted in the manuscript the text referring to grain diameter and the relative discussion.
The actual morphology must be confirmed or updated. In case the morphology is confirmed the shape (triangular) of grains have to be explicitly mentioned and discussed. Accordingly, also numerical data stemming from analysis have to be confirmed or updated.
- Authors show the XPS analysis and data selected “samples fabricated at 1200 °C, since the samples possess the lowest roughness in comparison with other samples”.
Why Authors used this criterion? Is XPS particularly sensitive to morphology in these samples? May the Authors explain?
- In the graphs, a spline curve is drawn connecting the data points: it may be misleading since it may recall a fitting curve due to a theoretical model. Please specify in the captions that it is only a guide for eyes. Or use a different graphical solution.
- In order to facilitate the readers, Authors are suggested should describe (or recall) before the discussion of Fig 11 how they obtained the square areas with different current and potential properties (i.e. the two scans at different size, scanned at -3V and +3V).

Author Response
Response to Reviewer 1 Comments
Point 1: The Figure 1 shows AFM images of ZnO samples morphology. What is apparent is the probable presence of artefacts in the images due to the worn tip. More specifically the features that indicate the presence of artifacts in the images are the triangular features with the same orientation which are present in all the different images (in one of them the features has a trapezoidal shape with same orientation of the borders). The Authors should check if by rotating the sample respect to the cantilever the features remain the same (artifacts) or rotates (actual features of the sample morphology). If those reported in the manuscript are not the actual sample morphologies the authors should show (replace) additional data and discuss the experimental findings accordingly. It is likely that the trends observed in roughness and grain diameter are actual, but the absolute may be different.
As a consequence data and discussion which refers to the grain diameter which appear in several parts of the manuscript may require amendment. I have highlighted in the manuscript the text referring to grain diameter and the relative discussion.
The actual morphology must be confirmed or updated. In case the morphology is confirmed the shape (triangular) of grains have to be explicitly mentioned and discussed. Accordingly, also numerical data stemming from analysis have to be confirmed or updated.
Response 1: Processing of experimental results of studying the dependence of grain diameter on annealing temperature was carried out on the basis of many AFM images, and images with artifacts were put in the article. Figure 1 has been updated to AFM images less susceptible to artifacts. The values grain diameter obtained during the analysis of many images remain valid in the article.
But even so we carried out a second study of ZnO films morphology, and found a discrepancy in the roughness values, which is most likely due to the lack of equipment calibration during the previous experiment. Thus the following changes were made:
Figure 1 was totally updated;
Figure 2a was updated;
Figure 5a was updated;
The sentence was added (lines 209-210):
An analysis of the results of measuring the ZnO/Al2O3 interface before annealing showed that the film thickness is 71.3±35.4 nm (Figure 1 f).
The sentence was added (lines 213-214):
It was found that with increasing annealing temperature, the surface roughness decreases from 35.0±4.1 nm to 13.1±1.8 nm, and the grain diameter increases from 231±12 nm to 511±35 nm.
The sentence was added (lines 385-387):
It was demonstrated that increasing the annealing temperature from 25 °Ð¡ to 1200 °Ð¡ leads to a decrease in the film roughness from 35.0±4.1 nm to 13.1±1.8 nm, and to an increase in grain size from 231±12 nm to 511±35 nm.
Point 2: - Authors show the XPS analysis and data selected “samples fabricated at 1200 °C, since the samples possess the lowest roughness in comparison with other samples”.
Why Authors used this criterion? Is XPS particularly sensitive to morphology in these samples? May the Authors explain?
Response 2: The choice of the sample with the smallest roughness is due to the fact that for the manufacture of ReRAM elements with reproducible parameters, one should strive to reduce the roughness of oxide films. This choice is not related to the XPS.
The sentence:
«To investigate the elemental composition and crystal structure, we selected samples fabricated at 1200 °C, since the samples possess the lowest roughness in comparison with other samples.»
has been explanded to
«To investigate the elemental composition and crystal structure, we selected samples fabricated at 1200 °C, since the samples possess the lowest roughness in comparison with other samples, which is important for the manufacture of ReRAM elements with reproducible parameters.»
(lines 220-222)
Point 3: In the graphs, a spline curve is drawn connecting the data points: it may be misleading since it may recall a fitting curve due to a theoretical model. Please specify in the captions that it is only a guide for eyes. Or use a different graphical solution.
Response 3:
Description of figure 2 changed to
Experimental investigation of annealing temperature effect on surface roughness (a) and grain’s diameter (b).
Description of figure 4 changed to
Experimental investigation of annealing temperature effect on electron concentration (a), electron mobility (b), and resistivity (c).
Description of figure 5 changed to
Experimental investigation of annealing time effect on surface roughness (a) and resistivity (b) at different substrate temperatures and chamber pressures.
Description of figure 6 changed to
Experimental investigation of electroforming voltage effect on grain diameter.
Description of figure 8 changed to
Experimental investigation of grain diameter effect on: (a) – nanocrystalline ZnO film resistance; (b) – RHRS/RLRS ratio.
Description of figure 9 changed to
Experimental investigations of RHRS/RLRS ratio dependences on electroforming voltage (a) and electroforming time (b) at various grain diameters.
Description of figure 10 changed to
Experimental investigations of set voltage dependences on electroforming voltage (a) and electroforming time (b) at various grain diameters.
Description of figure 13 changed to
Experimental investigations of dependences of RHRS/RLRS ratio on (a) current compliance and (b) sweep voltage.
Point 4: - In order to facilitate the readers, Authors are suggested should describe (or recall) before the discussion of Fig 11 how they obtained the square areas with different current and potential properties (i.e. the two scans at different size, scanned at -3V and +3V).
Response 4:
The sentence:
«The analysis of the AFM image of the ZnO film (Figure 11a) and the spreading resistance map (Figure 11b) shows the presence on the surface of a region in the HRS state with a resistance of (8.3±1.2)×10-9 Ω (dark contrast), and region in the LRS state with resistance (2.1±0.6)×10-9 Ω (light contrast).»
has been replaced by sentence
«An analysis of the 10x10 μm2 and 6x6 μm2 regions obtained by conductive AFM on the ZnO film surface (Figure 11) at voltages of -3 and 3 V, respectively, showed the presence on the surface of a region in the HRS state with a resistance of (8.3±1.2)×10-9 Ω (dark contrast), and region in the LRS state with resistance (2.1±0.6)×10-9 Ω (light contrast).»
(lines 341-344)
Point 5: (PDF comment) AFM should be close to the model. "....AFM in the semi contact mode..." .
Please report the spring constant for contact afm, and/or working frequency according to the different AFM mode used
Response 5: The sentence:
«The morphology of ZnO nanocrystalline films (Figure 1) was studied using the Ntegra Probe Nanolaboratory (NT-MDT, Russia) in the semi-contact AFM mode using NSG11 cantilevers.»
has been replaced by sentence
«The morphology of ZnO nanocrystalline films (Figure 1) was studied by AFM in the semi-contact mode using the Ntegra Probe Nanolaboratory (NT-MDT, Russia) and a commercial cantilever an NSG11 with 255 kHz resonant frequency and 11.8 N/m spring constant.»
(lines 136-138)
Expression
«For this purpose, ETALON HA_HR cantilevers with a conductive coating of W2C were used.»
has been replaced by
«For this purpose, a commercial ETALON HA_HR cantilevers with a conductive coating of W2C, 380 kHz resonant frequency and 34 N/m spring constant were used. »
(lines 140-142)
Point 6: (PDF comment) "Conductive AFM" is more frequently.
Response 6: Expression «Current AFM» has been replaced by « Conductive AFM » (lines 140, 190)
Point 7: (PDF comment) Please specify energy resolution in the spectra with the acquisition parameters used
Response 7: Description of the technique has been expanded:
The structure of ZnO films was studied by X-ray photoelectron spectroscopy using an ESCALAB 250Xi spectrometer (Thermo Scientific, USA) combined spectrometer with monochromatization of the Al Kα X-ray radiation line. The energy resolution was determined in reference to the Ag 3d5/2 line and corresponded to 0.6 eV. In the study, the spatial resolution was 250 μm. Also the structure of ZnO films was studied using X-ray diffractometry using a Rigaku Miniflex 600 diffractometer (Rigaku Corporation, Japan) (Figure 3).
(lines 144-149)

Reviewer 2 Report
Although many articles have been published on the electrical properties of thin nanocrystalline ZnO films, the contribution „Synthesis and memristor effect of a forming-free ZnO nanocrystalline films“ deals with interesting topic of memristors.
The introduction to ReRAM systems and memristors are elaborated at a high level. Nevertheless I am missing the information about the ALD and ZnO films, which are nice review in Laurenti M. et all in publication” Zinc Oxide Thin Films for Memristive Devices: A Review(Review) in critical Reviews in Solid State and Materials Sciences, 42, 2017, Pages 153-172.
In second paragraph Materials and Methods I am missing few important details and their compliance with the information provided in this paper. There is not given the thickness of the deposited films after deposition. Only in page 6 is written 64 nm for the films with XRD in Fig. 3b.
On page 3: As a result, 5 samples Al2O3/TiN/ZnO were prepared, which were annealed in a nitrogen atmosphere with a pressure of 10-3 Torr at annealing temperatures of 25 °C (i.e. without annealing), 600 °C, 800 °C, 1000 °C, and 1200 °C for 10 hours each. But in the Results and discussion I found also annealing on 300 °C, at different nitrogen pressure and different time. Please summarised all experimental details more carefully.
Did the authors checked if the thickness of the films is not varied by the annealing in nitrogen ?
So the first part of this paragraph on page 3 has to be rewrite and brought into line with what is given in the rest of publication.
Page 6: XPS. I don’t understand the sentence : “In this case the concentration of oxygen vacancies decreases due to chemisorption, which allows oxygen to interact with a sufficient number of Zn atoms to form ZnO [54].“ Because you are annealing in nitrogen, so it has to increase the oxygen vacancies. Moreover for readers will be more important to see the peaks corresponding to O1s and Zn 2p as well as N1s in more detail instead of survey. It could be given in supplement. In Fig. 3b is shown XRD, but there is no information before or after annealing . If the XRD was performed the information about the crystalline size (Scherer`s formula or Williamso-Hall plot) could be interesting information in related to the grain size obtained by AFM.
Because for the electrical measurements are more important the structure perpendicular to the film the cross section SEM or TEM could bring interesting information about the grain size and structure, which cannot be received only from AFM. At least for few samples these analyses could be included to the paper.
May be the size of the upper tungstate electrode could be mentioned.
The part concerning electrical measurements and explanation is very interesting and inspiring for materials research community
Author Response
Response to Reviewer Comments
Point 1: The introduction to ReRAM systems and memristors are elaborated at a high level. Nevertheless I am missing the information about the ALD and ZnO films, which are nice review in Laurenti M. et all in publication” Zinc Oxide Thin Films for Memristive Devices: A Review(Review) in critical Reviews in Solid State and Materials Sciences, 42, 2017, Pages 153-172.
Response 1: The sentence was added by phrase «atomic layer deposition (ALD) [68]» (lines 110-113):
Currently, the following technological methods are widely used in the fabrication of ZnO nanocrystalline films: magnetron sputtering [63], chemical vapor deposition [64], sol-gel process [65], anodic oxidation [66], thermal evaporation [67], atomic layer deposition (ALD) [68], and pulsed laser deposition (PLD) [69, 70].
References were extended (lines 601-602):
- 68. Laurenti, M.; Porro, S.; Pirri, C. F.; Ricciardi, C.; Chiolerio, A. Zinc oxide thin films for memristive devices: a review. Critical Reviews in Solid State and Materials Sciences 2017 42(2), 153-172. [CrossRef]
Point 2: In second paragraph Materials and Methods I am missing few important details and their compliance with the information provided in this paper. There is not given the thickness of the deposited films after deposition. Only in page 6 is written 64 nm for the films with XRD in Fig. 3b.
Response 2: The sentences was added (lines 138-139):
«The ZnO thickness was determined from the ZnO/Al2O3 interface measuring.»
and (lines 209-210):
«An analysis of the results of measuring the ZnO/Al2O3 interface before annealing showed that the film thickness is 71.3±35.4 nm (Figure 1 f).»
Cross-section was added to Figure 1
The sentence (line 231):
«Figure 3b shows the XRD spectrum of a ZnO film which had a thickness of 64 nm»
was changed to
«Figure 3b shows the XRD spectrum of a ZnO film without annealing »
Point 3: On page 3: As a result, 5 samples Al2O3/TiN/ZnO were prepared, which were annealed in a nitrogen atmosphere with a pressure of 10-3 Torr at annealing temperatures of 25 °C (i.e. without annealing), 600 °C, 800 °C, 1000 °C, and 1200 °C for 10 hours each. But in the Results and discussion I found also annealing on 300 °C, at different nitrogen pressure and different time. Please summarised all experimental details more carefully.
Response 3: Here we mean, that 25 °C (i.e. without annealing), 600 °C, 800 °C, 1000 °C, and 1200 °C – substrate temperatures during annealing, but 300 °C and 800 °C ‑ substrate temperatures during ZnO deposition.
Lines 154-155: «To study the effect of annealing time on roughness and resistivity, 4 samples of Al2O3/TiN/ZnO were fabricated, obtained at different substrate temperatures of 300 °C and 800 °C.»
Point 4: Did the authors checked if the thickness of the films is not varied by the annealing in nitrogen ?
Response 4: This experiment has not been carried out.
Point 5: Page 6: XPS. I don’t understand the sentence : “In this case the concentration of oxygen vacancies decreases due to chemisorption, which allows oxygen to interact with a sufficient number of Zn atoms to form ZnO [54].“ Because you are annealing in nitrogen, so it has to increase the oxygen vacancies. Moreover for readers will be more important to see the peaks corresponding to O1s and Zn 2p as well as N1s in more detail instead of survey. It could be given in supplement. In Fig. 3b is shown XRD, but there is no information before or after annealing . If the XRD was performed the information about the crystalline size (Scherer`s formula or Williamso-Hall plot) could be interesting information in related to the grain size obtained by AFM.
Response 5:
The sentence
«In this case the concentration of oxygen vacancies decreases due to chemisorption, which allows oxygen to interact with a sufficient number of Zn atoms to form ZnO [54]»
is related to oxygen annealing, mentioned in the previous sentence.
Figure 3b shows the XRD spectrum of a ZnO film without annealing (line 231)
Point 6:
Because for the electrical measurements are more important the structure perpendicular to the film the cross section SEM or TEM could bring interesting information about the grain size and structure, which cannot be received only from AFM. At least for few samples these analyses could be included to the paper.
Response 6:
This experiment will be carried out for our future articles; in this article, the authors did not aim to study the structure of the films before and after switching using SEM or TEM.
Point 7: May be the size of the upper tungstate electrode could be mentioned.
Response 7: The sentence (lines 164-165):
«The TiN film served as the lower contact; a tungsten probe was used as the upper contact.»
was extended to:
«The TiN film served as the lower contact; a tungsten probe with diameter 150 nm was used as the upper contact.»

Reviewer 3 Report
In this paper, the authors investigated synthesis and memristor effect of a forming-free ZnO nanocrystalline films. Some information in this paper is useful, and the contents seems correct and interesting for some readers. I will give the following comments to improve this paper further.
In "1.Introduction", the authors wrote about "neuromorphic systems" in detail, which is an important application of memeristors, including your memristor using ZnO nanocrystalline films. But, in the body text, there is no explanation how to implement your memristor into neuromorphic systems. I recommend you to add some explanation how to implement your memristor into neuromorphic systems, otherwise at least to cite prior references related to the implementation method of memristors into neuromorphic systems.
Author Response
Response to Reviewer Comments
Point 1: In "1.Introduction", the authors wrote about "neuromorphic systems" in detail, which is an important application of memeristors, including your memristor using ZnO nanocrystalline films. But, in the body text, there is no explanation how to implement your memristor into neuromorphic systems. I recommend you to add some explanation how to implement your memristor into neuromorphic systems, otherwise at least to cite prior references related to the implementation method of memristors into neuromorphic systems.
Response 1: Paragraph was added (lines 374-381):
As a result, we determined the regimes of nanocrystalline ZnO films manufacturing exhibiting a stable memristive effect with a RHRS/RLRS ratio 78.34±5.31 and a switching voltage USET 1.9±0.2 V. The absence of the need for an electroforming allows one to avoid a number of problems with the ReRAM element efficiency, associated with information loss due to reducing the weight of synapses between neurons, and also allows us to increase the output of workable ReRAM elements. Thus, the results of the study, in conjunction with the results of other scientific groups studying the use of zinc oxide for biological synapses [74-76], show us the prospects of using ZnO nanocrystalline films for neuromorphic systems manufacturing.
References were extended (lines 618-627):
- Wang, D. T.; Dai, Y. W.; Xu, J.; Chen, L.; Sun, Q. Q.; Zhou, P.; Zhang, D. W. Resistive switching and synaptic behaviors of TaN/Al 2 O 3/ZnO/ITO flexible devices with embedded Ag nanoparticles. IEEE electron device letters 2016, 37(7), 878-881. [CrossRef]
- Dang, B.; Wu, Q.; Song, F.; Sun, J.; Yang, M.; Ma, X.; Hao, Y. A bio-inspired physically transient/biodegradable synapse for security neuromorphic computing based on memristors. Nanoscale 2018, 10(43), 20089-20095. [CrossRef]
- Dongale, T. D.; Mullani, N. B.; Patil, V. B.; Tikke, R. S.; Pawar, P. S.; Mohite, S. V.; Shinde, S. S. Mimicking the biological synapse functions of analog memory, synaptic weights, and forgetting with ZnO-based memristive devices. Journal of Nanoscience and Nanotechnology 2018, 18(11), 7758-7766. [CrossRef]

Round 2
Reviewer 1 Report
Dear Authors,
I have found satisfactory responses to my comments, and also my requests were adequately fulfilled.
Regards
Reviewer 2 Report
The paper was suficiently improved and is suitable for publication in present form.